

# Mechanisms and potential immune tradeoffs of accelerated coral growth induced by microfragmentation

Louis Schlecker[1], Christopher Page[2], Mikhail Matz[3] and
Rachel M. Wright[1,3]

[1] Smith College, Northampton, Massachusetts, United States
[2] Mote Marine Laboratory, Summerland Key, Florida, USA
[3] University of Texas at Austin, Austin, Texas, United States

## ABSTRACT

Microfragmentation is the act of cutting corals into small pieces (~1 cm$^2$) to accelerate the growth rates of corals relative to growth rates observed when maintaining larger-sized fragments. This rapid tissue and skeletal expansion technique offers great potential for supporting reef restoration, yet the biological processes and tradeoffs involved in microfragmentation-mediated accelerated growth are not well understood. Here we compared growth rates across a range of successively smaller fragment sizes in multiple genets of reef-building corals, *Orbicella faveolata* and *Montastraea cavernosa*. Our results confirm prior findings that smaller initial sizes confer accelerated growth after four months of recovery in a raceway. *O. faveolata* transcript levels associated with growth rate include genes encoding carbonic anhydrase and glutamic acid-rich proteins, which have been previously implicated in coral biomineralization, as well as a number of unannotated transcripts that warrant further characterization. Innate immunity enzyme activity assays and gene expression results suggest a potential tradeoff between growth rate after microfragmentation and immune investment. Microfragmentation-based restoration practices have had great success on Caribbean reefs, despite widespread mortality among wild corals due to infectious diseases. Future studies should continue to examine potential immune tradeoffs throughout the microfragmentation recovery period that may affect growout survival and disease transmission after outplanting.

## INTRODUCTION

Reefs occupy less than 1% of the ocean area, but the physical structures produced by calcifying reef corals support roughly 34% of the ocean's biodiversity (*Reaka-Kudla, 2001*). A variety of environmental factors threaten this biodiversity as coral ecosystems rapidly decline (*Hoegh-Guldberg et al., 2007*; *Sully et al., 2019*; *Dietzel et al., 2020*). The Caribbean has experienced 50–80% reductions in coral cover in the last few decades due to

Corresponding author
Rachel M. Wright,
rwright@smith.edu

increasingly frequent and intense bleaching events, hurricanes, and disease outbreaks (*Aronson & Precht, 2001*; *Gardner et al., 2005*; *Hoegh-Guldberg et al., 2007*; *Miller et al., 2009*, *Sully et al., 2019*). Coral disease was rarely identified before 1980, but has become more prevalent with rising temperatures (*Bruno et al., 2007*; *Harvell et al., 2007*). There are now over 40 recognized coral diseases affecting roughly 200 coral species in over 75 countries (*Bruckner, 2016*). Some areas of Florida's reefs have been reduced to less than 3% of their previously recorded population densities in part due to disease, which is a consequence of complex interactions between biotic (e.g., pathogen abundance) and abiotic (e.g., temperature) factors (*Randall et al., 2014*; *Precht et al., 2016*). However, some reports suggest these declines may have begun before disease outbreaks became prominent (*Cramer et al., 2020*). Stony Coral Tissue Loss Disease (SCTLD) was first reported off the southeast coast of Florida in 2014 after a major bleaching event and has since spread quickly throughout the Caribbean (*Weil et al., 2019*), further contributing to Caribbean coral loss (*Meiling et al., 2021*). The etiology of SCTLD is currently unknown, though some bacterial taxa are strongly associated with pathogenesis (*Rosales et al., 2020*; *Landsberg et al., 2020*), and multiple studies have shown that an interventive antibiotic treatment can slow lesion progression (*Aeby et al., 2019*; *Neely et al., 2020*; *Shilling, Combs & Voss, 2021*). These declines in coral cover demonstrate that active restoration efforts are increasingly necessary to help prevent the loss of coral ecosystems in the Caribbean, though gaps in restoration science can limit the effectiveness of management practices (*Boström-Einarsson et al., 2020*).

One coral restoration technique is microfragmentation: a process in which corals are divided into very small pieces (usually 1–3 cm$^2$) that then grow rapidly to generate coral biomass (*Forsman et al., 2015*; *Page, Muller & Vaughan, 2018*). Coral fragments can be directly outplanted or transferred to a land-based or *in situ* nursery to grow for 6–12 months before being outplanted to a recipient reef (*Page, Muller & Vaughan, 2018*). Microfragmentation offers clear benefits for quickly generating coral biomass, but much remains unknown about the biological processes and potential consequences of this rapid growth. Previous work demonstrates that variation in growth rate among calcifying corals can be a poor predictor of overall fitness (*Edmunds, 2017*). Like all living organisms, a coral's energetic budget simultaneously supports all aspects of growth and homeostasis (*Lesser, 2013*), including immune activity to prevent disease (*Palmer, 2018*) and lesion repair to heal physical wounds (*Denis et al., 2011*). Corals experiencing environmental challenges on today's reefs will not survive if they cannot sustain rapid growth along with robust responses to biotic and abiotic threats.

Corals rely on innate immunity to ward off infection by invading pathogens (*Palmer & Traylor-Knowles, 2012*). Key players of these immune activities include phenoloxidase, an enzyme involved in the melanin synthesis pathway (*Mydlarz & Palmer, 2011*), and antioxidant enzymes catalase and peroxidase (*Palmer et al., 2011*). Melanization contributes to coral disease resistance and clearance by concentrating cytotoxic melanin around invading pathogens and damaged tissue (*González-Santoyo & Córdoba-Aguilar, 2012*). The melanization process and other host responses to pathogens produce reactive oxygen species as byproducts that provide additional antimicrobial defenses (*Cerenius, Lee*

& *Söderhäll, 2008*), but can potentially damage the host's healthy tissue (*Sadd & Siva-Jothy, 2006*). Thus, antioxidant enzymes such as peroxidase and catalase are also integral to immune activity (*Mydlarz, McGinty & Harvell, 2010*). Coral fluorescent proteins may also provide peroxide scavenging properties that aid immunity (*Palmer, Modi & Mydlarz, 2009*).

Gene expression analyses provide molecular insight to stress responses in corals. For example, peroxidase expression has been associated with coral responses to thermal stress and disease (*Mydlarz & Harvell, 2007*; *Wright et al., 2017*; *Traylor-Knowles et al., 2021*). Gene expression analyses have also highlighted the importance of other cellular mechanisms to manage disease, such as programmed cell death. *O. faveolata* and *M. cavernosa* upregulated apoptosis-related genes when exposed to SCTLD (*Traylor-Knowles et al., 2021*). Corals lack the adaptive immunity required to recognize specific pathogens from previous exposures, but they do possess receptors that recognize general pathogen-associated molecular patterns, such as Toll-Like Receptors (TLRs) (reviewed in *Nie et al., 2018*). TLRs have been characterized in *O. faveolata* (*Williams et al., 2018*) and were upregulated in *Acropora hyacinthus* affected with white syndromes (*Wright et al., 2015*). Antimicrobial peptides (AMPs) provide another innate immune mechanism to restrict bacterial infections by directly targeting bacterial cells and by initiating TLR immune pathways (*Lee, Lee & Wong, 2019*). Differential expression of AMPs in response to *Vibrio* challenge has been documented in the reef-building coral *Pocillopora damicornis* (*Vidal-Dupiol et al., 2011*). Lectins are another type of bioactive peptide that can recognize non-self molecular patterns to activate immune mechanisms in corals (*Palmer & Traylor-Knowles, 2012*). C-type lectin was downregulated in *O. faveolata* affected with White Plague Disease relative to healthy corals, suggesting a diminished capacity for affected corals to respond to bacterial infection (*Daniels et al., 2015*).

The cellular machinery driving calcification, immune defense, and tissue regeneration requires energetic investment. The present study implemented variable size fragmentation that induced different growth rates to study the biological underpinnings of and potential tradeoffs associated with rapid growth in reef-building corals *O. faveolata* and *M. cavernosa*. The association between innate immune enzyme levels and initial fragment size indicates the relationship between investment in growth and immunity. Transcriptomic analysis by initial fragment size (i.e., the extent of microfragmentation) and buoyant weight change identified genes associated with microfragmentation and calcification.

## MATERIALS AND METHODS

### Coral collection, microfragmentation, and growth

Large fragments (~140–370 cm$^2$) of five colonies from each species (*M. cavernosa* and *O. faveolata*) were collected from the NOAA Rescue Nursery in Key West, a repository for stony corals which were rescued from construction sites, in November 2015 under Florida Keys National Marine Sanctuary (FKNMS) permit FKNMS-2015-130. These large colonies were first processed on 10 Nov 2015 using a seawater-cooled tile saw (MK 101 Pro Series; MK Diamond Products Inc., Torrance, CA, USA) to remove the majority of the
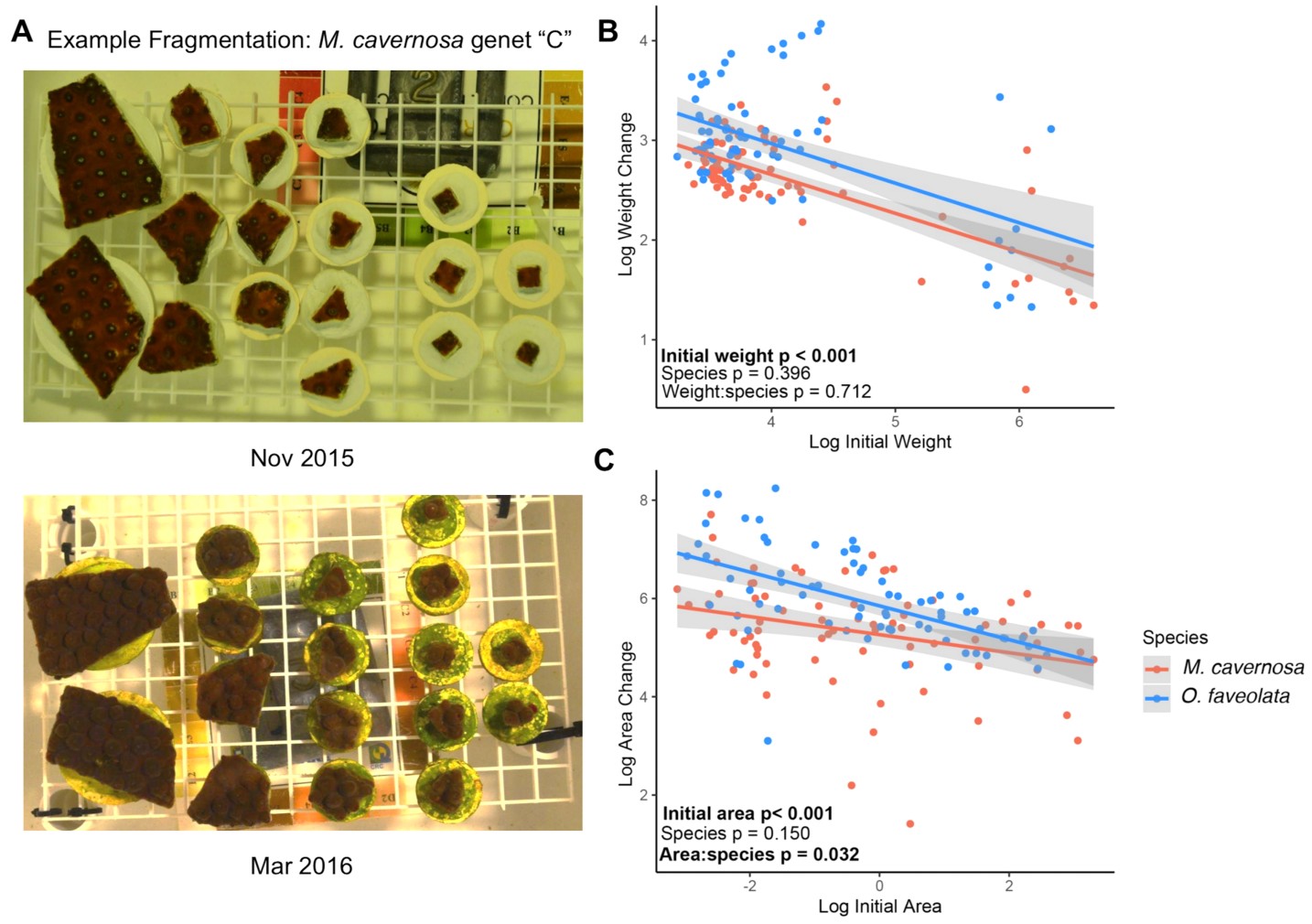

**Figure 1 Smaller fragments grow faster.** (A) Example photographs of coral fragments from a single *M. cavernosa* genet "C" taken in Nov 2015 (top) and after the 4-month recovery period in March 2016 (bottom). (B) Relationships between log-transformed initial weight and change in buoyant weight and log-transformed initial area and change in surface area (C) for *M. cavernosa* (red) and *O. faveolata* (blue). Each point represents an individual coral fragment (B and C). Shaded lines represent 95% confidence intervals for the linear regression between initial size and growth. *P*-values represent results from the MCMC model testing the individual effects of initial size and species, and the interaction between initial size and species. Significant values are in bold.

dead skeleton from the base of each colony. Colonies were then cut into a range of fragment sizes from about 0.1–10 cm$^2$ using a seawater cooled diamond band saw (C-40; Gryphon Corporation, Foothill Boulevard, CA, USA) and mounted to cement plugs using underwater epoxy (Allfix; Cir Cut Corporation, Philadelphia, PA, USA). Each fragment was photographed with a color and size standard (example in Fig. 1A) and weighed following the buoyant weight protocol (*Spencer Davies, 1989*). Surface areas were measured using ImageJ (*Schneider, Rasband & Eliceiri, 2012*).

Fragments were then placed in a 340 L raceway supplied with ~35 ppt seawater from a 24 m deep well. Seawater exiting the well was degassed of $CO_2$ and filtered mechanically using a sand filter and 100 μm pleated filter (Pentair Aquatic Ecosystems Inc., Apopka, FL, USA) to bring the pH to 7.7 before entering the raceway at a rate of

2.5 Lpm. The raceway was supplied with four 3 cm airstones (Sweetwater; Pentair Aquatic Ecosystems, Apopka, FL, USA) to further degas incoming seawater to a pH of 8.0 and to provide circulation. Additionally, the raceway was housed under a metal canopy lined with 40% shade cloth, and later draped with an additional shade cloth at 14:00 to prevent over irradiance, such that the photosynthetically active radiation during the day ranged from ~60–700 $\mu$mol m$^{-2}$s$^{-2}$, peaking during midday (Model QMSS-E; Apogee Instruments Inc., Logan, UT, USA). Daily siphoning, grazing by the sea snail *Lithopoma tecta*, and periodic manual removal of long chain diatoms kept fragments free of invasive algae during the experiment. Fragments were photographed and weighed again on 12 March 2016 using the same measurement protocols as previously described. After weighing, a small (~1 cm$^2$) amount of tissue was preserved in ethanol for gene expression analysis. The remaining tissue was snap-frozen on dry ice and preserved at −80 °C for protein analysis.

We note that some surface areas of the largest fragments may be underestimated due to limitations in the photographic method used for measurements. Some of the larger fragments had grown over and down the side of the cement pucks, out of the field of view from the top-down photographs (example images in Fig. 1A). Thus, we focus on growth in terms of weight captured by the sensitive buoyant weight method, which represents skeletal growth that is deposited by living tissue, for the subsequent associations between growth and immune activity or gene expression.

## Immune activity assays

Coral protein was extracted following previously described protocols (*Wright et al., 2017*). Briefly, coral tissue was removed using an airbrush and cold (4 °C) extraction buffer (100 mM Tris-HCl, pH 7.8, with 0.05 mM dithiothreitol). The resulting tissue slurries were homogenized by vortexing samples with a small amount of 1 mm glass beads (BioSpec, Bartlesville, OK, USA) for two minutes. The homogenized tissue slurry was centrifuged at 4 °C for 10 min at 3,200 g to separate coral and algal fractions. The coral protein supernatant (protein extraction) was removed and stored at −80 °C until use. Surface area determinations of airbrushed skeletons were made using the foil technique (*Marsh, 1970*). Briefly, aluminum foil was carefully trimmed along the area of the coral skeleton where tissue was removed. The weight of the trimmed foil was compared to a standard curve of known surface areas to estimate the surface area.

Total protein was assessed in triplicate using the RED660 protein assay (G Biosciences, St. Louis, MO, USA) with a bovine serum albumin standard curve. Sample absorbance, measured at 660 nm using a SpectraMax plate reader (Molecular Devices), was compared to the standard curve and normalized to tissue surface area and slurry volume to account for dilution with extraction buffer.

Immune enzyme activities were measured in triplicate as previously described (*Wright et al., 2017*). Active phenoloxidase (PO) activity was measured by mixing 20 $\mu$L of sodium phosphate buffer (50 mM, pH 7.0), 25 $\mu$L of sterile water, and 20 $\mu$L of protein extract. Dopamine (30 $\mu$L, 10 mM) was added as substrate and absorbance at 490 nm was measured every 30 s for 15 min. Change in absorbance was calculated during the linear

range of the curve (1–3 min). Activity was expressed as the change in absorbance per mg of protein ($\Delta A490 \cdot$ mg protein$^{-1} \cdot$ min$^{-1}$). Total phenoloxidase activity (PPO), including the inactive prophenoloxidase and active PO, was measured the same as PO except for the addition of 25 µL of trypsin (0.1 mg $\cdot$ mL$^{-1}$) in the reaction buffer to activate prophenoloxidase. Catalase (CAT) activity was measured by mixing 45 µL of sodium phosphate buffer (50 mM, pH 7.0), 75 µL of 25 mM $H_2O_2$, and 5 µL of protein extract. Samples were loaded on ultraviolet transparent plates (UltraCruz; Santa Cruz Biotechnology, Dallas, TX, USA) and absorbance at 240 nm was measured every 30 s for 15 min. Change in absorbance was calculated during the linear range of the curve (1–3 min). Activity was expressed as the change in hydrogen peroxide concentration per mg of protein ($\Delta H_2O_2 \cdot$ mg protein$^{-1} \cdot$ min$^{-1}$). Peroxidase (POX) activity was measured by mixing 40 µL of sodium phosphate buffer (10 mM, pH 6.0), 25 µL of 10 mM guaiacol, and 10 µL of protein extract. Absorbance at 470 nm was measured every 30 s for 15 min. Change in absorbance was calculated during the linear range of the curve (1–3 min). Activity was expressed as the change in absorbance per mg of protein ($\Delta A470 \cdot$ mg protein$^{-1} \cdot$ min$^{-1}$).

## Tag-Seq library preparation, sequencing, and analysis

A small (<1cm$^2$) amount of tissue was removed from the growing edge of each coral fragment and immediately preserved in cold ethanol before transfer to −80 °C. The remaining coral tissue was preserved for protein analysis as previously described. RNA was isolated from each tissue sample using the RNAqueous Total RNA Isolation Kit (Invitrogen, Waltham, MA, USA). A total of 65 gene expression libraries, prepared following the TagSeq protocol (*Meyer, Aglyamova & Matz, 2011*), were of high enough quality for Illumina HiSeq 2500 sequencing (SRA: PRJNA764071). Reads were deduplicated, adapter sequences were trimmed, and low-quality reads (minimum quality score = 20; minimum percent bases above minimum quality score = 90%) were filtered using FASTX toolkit (*Hannon, 2010*). To determine dominant symbiont types for each species, we mapped TagSeq reads to a combined symbiont reference composed of transcriptomes from Symbiodiniaceae 'clades' A (Genus *Symbiodinium*) and B (Genus *Breviolum*) (*Bayer et al., 2012*) and 'clades' C (Genus *Cladocopium*) and D (Genus *Durusdinium*) (*Ladner, Barshis & Palumbi, 2012*) using a custom perl script 'zooxtype.pl'. Custom scripts for read deduplication and identifying Symbiodiniaceae genera are hosted within the 2bRAD GitHub repository (https://github.com/z0on/tag-based_RNAseq).

Trimmed TagSeq reads from *M. cavernosa* samples were mapped to a holobiont reference consisting of the *M. cavernosa* (Data & Code-Matz Lab), *Cladocopium goreaui* (*ReFuGe 2020 Consortium, 2015*), and *Durusdinium* genomes (*Shoguchi et al., 2021*) using Bowtie 2 (*Langmead & Salzberg, 2012*). Trimmed TagSeq reads from *O. faveolata* samples were mapped to a holobiont reference consisting of the *O. faveolata* (*Prada et al., 2016*) and *Durusdinium* genomes (*Shoguchi et al., 2021*). Reads were converted to counts representing the number of independent observations of a transcript over all isoforms for each gene. Significantly differentially expressed genes were characterized using NCBI BLAST (*Altschul et al., 1990*). Genes that returned no significant BLAST hit

were characterized using NCBI Conserved Domain Database (*Lu et al., 2020*) to predict functional protein domains.

Genes with a mean count of less than three across all samples were removed from the analysis, leaving 7889 genes for *M. cavernosa* and 9979 genes for *O. faveolata*. Read counts from technical replicates (libraries prepared from separate RNA extractions of the same coral fragment) were pooled before differential gene expression analysis.

Genotyping was performed with ANGSD v0.930 (*Korneliussen, Albrechtsen & Nielsen, 2014*) using coral reads mapped to their respective genomes as previously described. Sites were filtered to retain loci with a mapping quality ≥25 and minor allele frequency ≥0.05. Samples identified as clones in highly similar clusters based on distances among known clones (distance < 0.2) were re-identified as a single genet.

## Statistics

All statistical analyses were performed in R version 4.0.2 (*RStudio Team, 2020*). Growth was estimated as the percent change in buoyant weight ($\text{Weight}_{Final}$ − $\text{Weight}_{Initial}$/$\text{Weight}_{Initial}$ * 100) and as the percent increase in area ($\text{Area}_{Final}$ − $\text{Area}_{Initial}$/$\text{Area}_{Initial}$ * 100). We also estimated growth using the power of an exponential process, $\log_2$(final size/initial size), which gave the same results as percent growth estimates. Associations between growth and initial size were analyzed using Bayesian generalized linear mixed models implemented in MCMCglmm package in R (*Hadfield, 2010*) with the interaction between initial size and species as fixed factors and genet as a random effect.

Activities of CAT, POX, PO, and PPO were normalized to the total protein concentration, log-transformed, and compared among initial sizes using MCMCglmm. We used initial size as our predictor variable to address our biological objective to determine the relationship between immune parameters and the extent of microfragmentation (*i.e.*, initial fragment size). Log-transformation was chosen based on diagnostic plots of a linear model with species, genet, and initial fragment size as factors. The MCMC model included the interaction between initial size and species as fixed effects and genet as a random effect. Additionally, we conducted a binned analysis using categorical assignments for initial size as "small" (≤mean initial weight) or "big" (>mean initial weight).

Gene expression sample outliers were detected using arrayQualityMetrics (*Kauffmann, Gentleman & Huber, 2009*). Differentially expressed genes (DEGs) were identified using DESeq2 (*Love, Huber & Anders, 2014*). Wald tests were performed to compare continuous growth phenotypes using the models 'count ~ genet + initial weight + weight change' where genet was a factor representing the colony from which the fragment was made and weight values were continuous measurements that were centered and scaled per DESeq2 recommendations for continuous predictor variables to improve generalized linear model convergence. A total of 27 *M. cavernosa* and 25 *O. faveolata samples* remained after combining technical replicates and outlier detection. The DESeq2 models were run independently for each species. Count data was transformed using the variance stabilizing transformation. We reported Wald statistics (log fold change/standard error) to represent the magnitude of expression difference per unit change of continuous variables. We repeated DESeq2 gene expression analysis using binned categories of

Schlecker et al. (2022), *PeerJ*, DOI 10.7717/peerj.13158 

initial size and growth rate as previously described. False-discovery rate (FDR) p-values were adjusted using the Benjamini–Hochberg procedure (*Benjamini & Hochberg, 1995*). Gene expression heatmaps were generated using pheatmap (*Kolde, 2019*) and gene ontology enrichment was performed based on log-fold change values using GO-MWU (*Wright et al., 2015*). Permutational analysis of variance testing on Manhattan dissimilarity matrices (ADONIS) was performed using vegan (*Dixon, 2003*) to assess overall transcriptomic differences across samples. All analytical scripts and data files are available on GitHub (https://github.com/rachelwright8/Microfrag_Growth_Immunity_TagSeq).

A weighted gene correlation network analysis (WGCNA, *Langfelder & Horvath, 2008*) was used to correlate expression values for groups of co-regulated genes with traits. Genes with mean counts >3 were used to construct a signed network. A sample network was constructed to identify outlying samples with a standardized connectivity score of less than −2.5. A signed gene co-expression network was constructed with a soft threshold power of 12. Groups of co-regulated genes (modules) with a Pearson correlation coefficient 0.3 or higher were merged. GO enrichment for each module was performed as previously described, with the exception of using per-gene module membership values (kME) instead of log-fold change values.

## RESULTS

### Associations between growth and initial fragment size

Growth was monitored for 78 *M. cavernosa* and 74 *O. faveolata* microfragments as surface area expansion and as changes in buoyant weight, which included the weight of the cement plug. The mean ± SD values for initial area and weight across all *M. cavernosa* fragments were $1.72 \pm 2.36$ cm$^2$ and $22.8 \pm 21.8$ g, respectively. The mean ± SD values for initial area and weight across all *O. faveolata* fragments were $1.37 \pm 1.46$ cm$^2$ and $19.9 \pm 16.7$ g, respectively. After four months of growth, we observed a negative association between growth and initial fragment surface area (posterior mean = −0.17, $p < 0.001$) and between growth and buoyant weight (posterior mean = −0.30, $p < 0.001$) (Figs. 1B and 1C). There were no significant differences in mean growth between species. There was a significant interaction between species and surface area growth where *M. cavernosa* fragments displayed a less pronounced effect of microfragmentation on surface area growth than fragments of *O. faveolata* (posterior mean = −0.18; $p = 0.032$; Fig. 1C).

Plotting untransformed percent area increase for each fragment based on its initial size illustrates the range of microfragmentation-mediated growth effects (Fig. S1). We observed high variance in area increase below ~1 cm$^2$ ranging from tissue loss (−11%) to over 300% increase.

### Immune activity analyses

We obtained enough tissue with measurable levels of protein to conduct the immune activity assays from 47 *M. cavernosa* and 49 *O. faveolata* fragments (Figs. 2A–2D). We observed positive associations between initial fragment size and rates of PO (posterior mean = 1.74; $p = 0.012$), PPO (posterior mean = 2.55; $p = 0.006$), and CAT (posterior mean = 2.70; $p < 0.001$) activities. Only POX activity was not significantly associated with

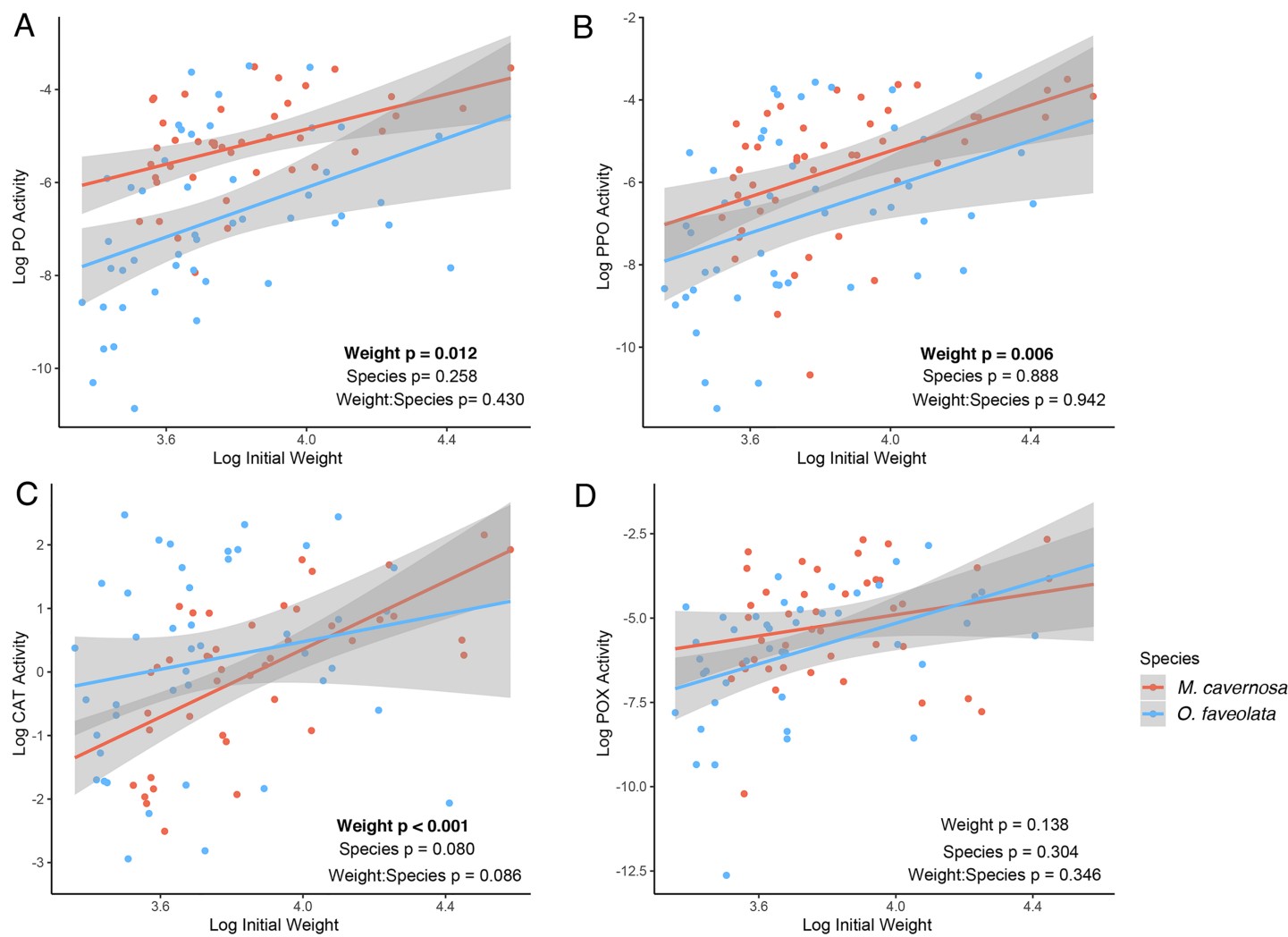

**Figure 2 Associations between log-transformed initial weight and log-transformed immune activities for active phenoloxidase (A), total phenoloxidase potential (B), catalase (C), and peroxidase (D) in *M. cavernosa* (red) and *O. faveolata* (blue).** Each point represents an individual coral fragment. Shaded lines represent 95% confidence intervals for the linear regression between initial size and growth. *P*-values represent results from the MCMC model testing the individual effects of and interaction between weight and species. Significant terms are indicated in bold.

coral size (posterior mean = 1.70; $p = 0.138$). There were no significant differences in immune activities between species. The binned analysis compared immune activities between fragments categorized as "big" (>mean initial weight, $N = 36$, mean ± SD = 17.5 ± 2.6 g) or "small" (≤mean initial weight, $N = 60$, mean ± SD = 12.3 ± 1.0 g). The binned analysis recapitulates the continuous analysis (Fig. S2). Larger fragments had higher PO (posterior mean = 0.97, $p = 0.012$), PPO (posterior mean = 1.37, $p = 0.010$), and CAT (posterior mean = 1.38, $p < 0.001$) activities than smaller fragments.

## Identifying clones and dominant symbiont types

Genetic distance cluster analysis across *O. faveolata* genets revealed that two genets (previously named "V" and "X") were clones, which were renamed as genet "U" in
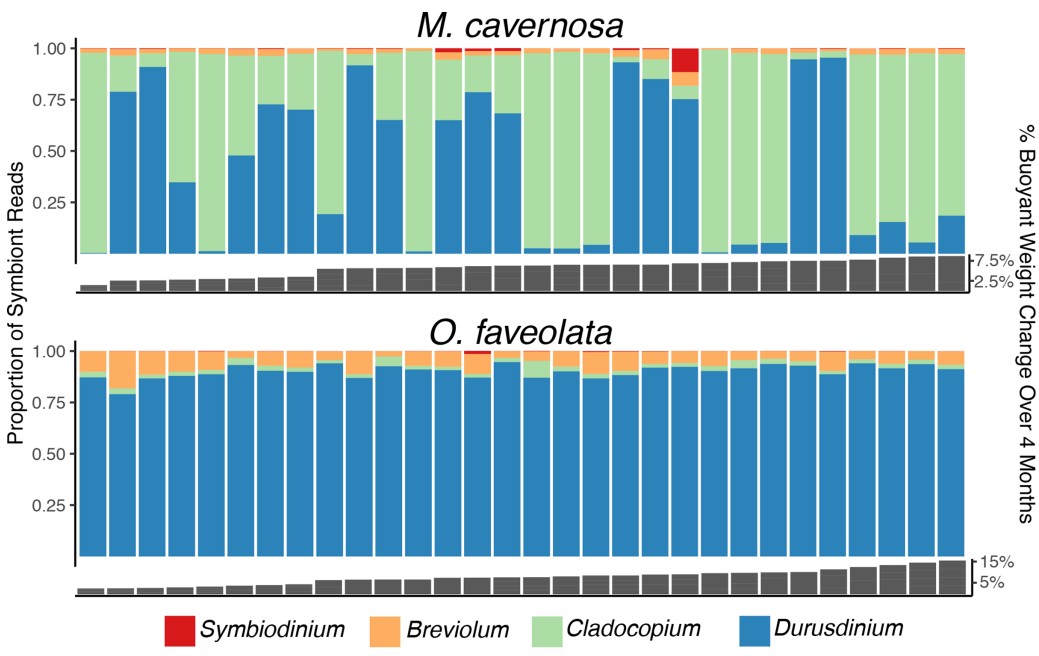

**Figure 3 Determination of dominant algal symbiont types based on the proportion of reads mapping to each reference for *M. cavernosa* (top) and *O. faveolata* (bottom).** Columns represent sequencing samples ordered by increasing buoyant weight percent increase (grey bars) from left to right. Colored bars represent the proportion of algal symbiont type reads according to the legend.

subsequent analysis (Fig. S3). Mapping *M. cavernosa* and *O. faveolata* reads to symbiont references determined that *M. cavernosa* hosted a mix of *Cladocopium* and *Durusdinium* while *O. faveolata* was dominated by *Durusdinium* (Fig. 3). We did not observe a significant association between growth (as percent surface area increase) and proportion *Durusdinium* reads in *M. cavernosa* samples (posterior mean = −50%, $p$ = 0.068), or when measuring growth as percent change in buoyant weight (posterior mean = −0.84, $p$ = 0.422) (Fig. S4).

## *M. cavernosa* host expression profiles with respect to initial size and growth

The average *M. cavernosa* holobiont mapping efficiency was 85.2% with per-sample averages of $1.83 \times 10^5$ trimmed and filtered reads mapping to the host genome. In the coral host, initial size ($p$ = 0.014, $r^2$ = 0.07) and genet ($p$ = 0.004, $r^2$ = 0.25) explained the majority of the observed differences in gene expression profiles (Fig. S5A). Weight change was not significantly associated with variation in gene expression (Fig. S5B; $p$ = 0.416). In the *M. cavernosa* holobiont analysis, only two host genes were significantly differentially expressed with respect to continuous growth rate at a threshold value of 0.1. These genes shared homology with transmembrane protein 86a (Mcavernosa16889) and guanylate binding protein (Mcavernosa20434). Transmembrane protein 86a expression had a positive association with growth (Wald stat = 4.35; FDR = 0.054) and guanylate binding

protein expression had a negative association with growth (Wald stat = −4.40; FDR = 0.054).

A binned gene expression analysis revealed 18 host genes that were significantly differentially expressed with respect to categorical initial fragment weight at FDR < 0.1 and 14 DEGs at FDR < 0.05 (Table S1). These genes include putative growth (*e.g.*, coadhesin) and immunity (*e.g.*, tachylectin-2) genes (Fig. S6). None of these 18 genes overlapped with the two DEGs identified using the continuous growth analysis.

We identified 28 significantly enriched (adjusted *p*-value < 0.1) gene ontology (GO) terms based on log-fold change values based on continuous weight change in the *M. cavernosa* host (Table S2). These enriched categories include terms related to protein synthesis and antioxidant responses. We did not identify any enriched GO terms for the host based on initial fragment size, nor did we find any enriched GO terms in the algal symbiont.

Using binned categories for initial fragment size and growth, we found four GO terms enriched with genes associated with initial fragment size and one GO term enriched with genes associated with growth (obsolete mitochondrial membrane part, delta rank = 444) (Table S2). The enriched terms regarding fragment size contained establishment of protein localization to membrane (delta rank = 468), ribosome (delta rank = 351), obsolete cytosolic part (delta rank = 385), and structural constituent of ribosome (delta rank = 417).

Several WGCNA modules represent potential expression tradeoffs between growth and immune metrics for *M. cavernosa* (Fig. 4). The green and violet modules were significantly positively associated with buoyant weight change and negatively correlated with immune parameters. The green module was enriched with genes related to structural constituents of ribosomes and mitotic cell cycle (Table S3), potentially reflecting genes associated with cellular growth. The dark magenta module was negatively associated with buoyant weight change and positively associated with immune parameters. This module had 52 significantly enriched GO terms, including "regulation of ossification," indicating the activity of genes that may be related to skeletal formation (*Scucchia et al., 2021*).

### *O. faveolata* host expression profiles with respect to initial size and growth

The *O. faveolata* holobiont mapping efficiency was 81.8% with per-sample average reads of $2.06 \times 10^5$ mapping to the host genome. In the coral host, differences between genets ($p = 0.001$, $r^2 = 0.35$) explained the majority of the observed differences in gene expression profiles (Fig. S7A). Initial weight and weight change were not significantly associated with variation in gene expression (Fig. S7B; $p = 0.139$ and 0.162, respectively).

We identified 38 host *O. faveolata* genes significantly associated with continuous weight change over the 4-month recovery period at a *p*-value threshold of 0.1 (Fig. 5). Of these genes, seven were positively associated with growth rate and 31 were negatively associated with growth rate. A gene sharing sequence homology with glutamic acid-rich protein (LOC110061392) had higher expression in faster growing corals (Wald stat = 3.57,

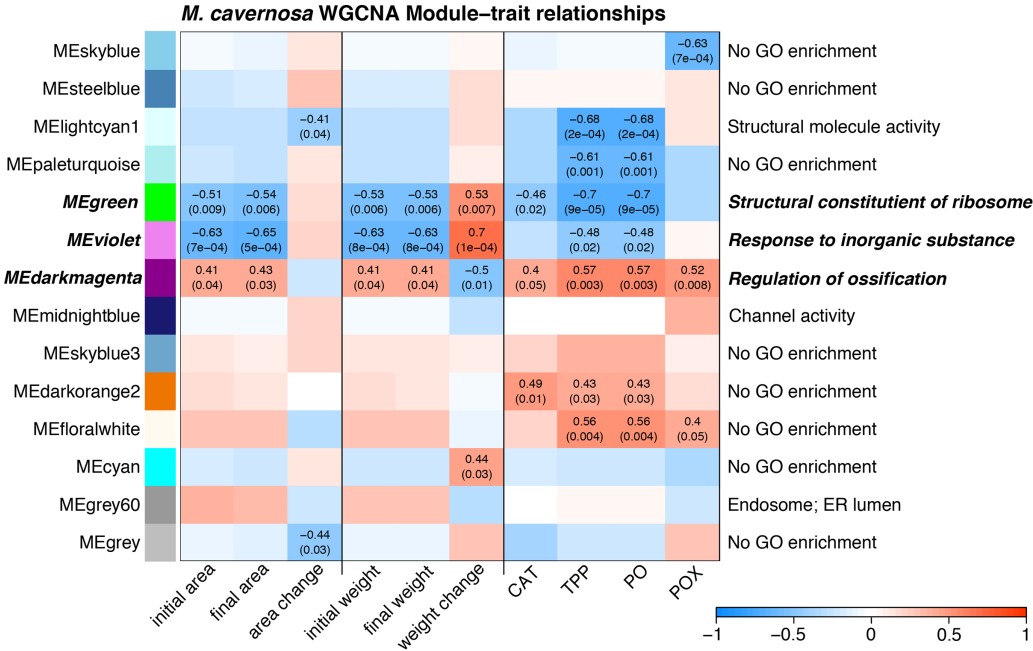

**Figure 4 Module–trait relationship heatmap for *M. cavernosa*.** The strength of the correlations between traits (terms indicated on along the x-axis) and gene coexpression modules (colored boxes along y-axis) are indicated by the intensity of color. Values within each cell indicate Pearson's correlation between the module eigengene and the trait and the *p*-value according to the correlation test for only significant correlations (*p* < 0.05). Terms along the right indicate a representative top enriched GO category for the module, if any. Modules in bold italic represent potential tradeoffs with opposite associations between growth and immune parameters.

FDR = 0.096). Two transcripts with homology to carbonic anhydrases had lower expression in faster growing corals (LOC110047361 Wald stat = −3.76, FDR = 0.07 & LOC110047363 Wald stat = −5.94, FDR = 2.8e−5). Six immune-related genes were negatively associated with growth. These genes include integumentary mucin C.1 (LOC110061602; Wald stat = −3.71, FDR = 0.083), a component of coral mucus (*Jatkar et al., 2010*), and cathepsin L1-like (LOC110068187; Wald stat = −3.59, FDR = 0.096), which has been linked to many immunological responses (*Brown & Rodriguez-Lanetty, 2015*). A gene with homology to green fluorescent protein (GFP)-like chromoprotein cFP484 (LOC110044329) was also negatively associated with growth (Wald stat = −4.11; FDR = 0.054).

We identified 10 host *O. faveolata* genes significantly associated with categorical weight change (small *vs* large) at a p-value threshold of 0.1 (Table S1). Three of these DEGs overlap with the results of the continuous analysis: 9-divinyl ether synthase-like, an uncharacterized gene containing a deoxycytidylate deaminase domain, and an uncharacterized gene with no predicted protein domains.

We identified three enriched GO terms regarding fragment size and three enriched GO terms regarding growth using log-fold change values associated with *O. faveolata* host genes in the continuous growth analysis (Table S2). GO terms related to fragment size

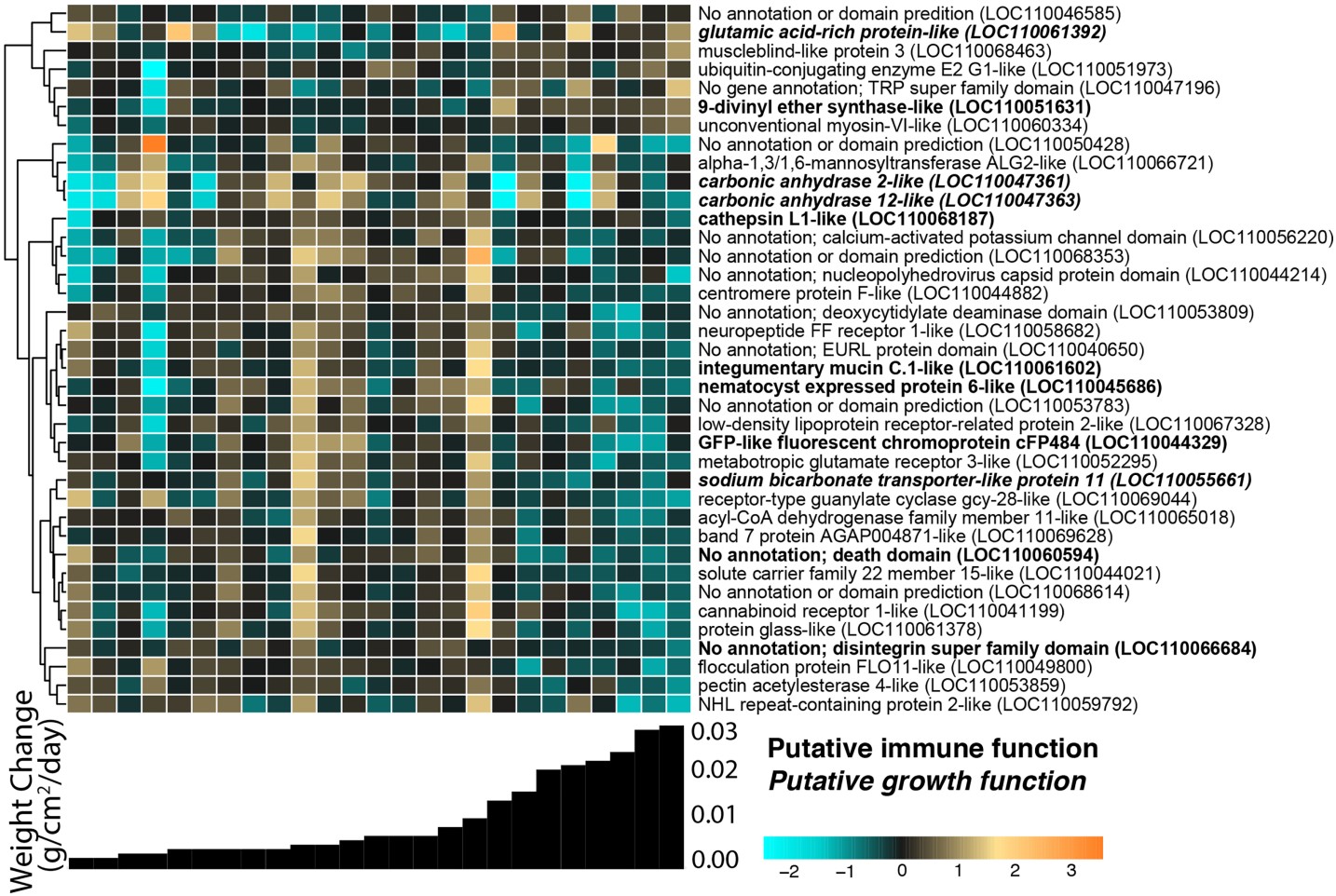

**Figure 5** *O. faveolata* **gene expression differences across samples with different growth rates.** Rows are genes and columns are samples. The color scale indicates log2-fold change relative to the mean expression of each gene across all samples. Genes are hierarchically clustered based on Pearson's correlations of expression across samples. Bar graphs below each column represent the per-day growth rate as weight change per cm² over the 4-month recovery period for that sample. Genes with a putative immune function are highlighted in bold; genes with a putative growth function are bolded and italicized.

contained transmembrane signaling receptor activity (delta rank = −290), G protein-coupled receptor activity (delta rank = −349), and molecular transducer activity (delta rank = −221). GO terms relating to growth contained polytene chromosome (delta rank = 301), exocytic vesicle (delta rank = 309), and transmembrane signaling receptor activity (delta rank = −241). We found no enriched GO terms using binned categories for initial fragment size and growth.

Several WGCNA modules were correlated with immune activity in *O. faveolata*, but these results should be considered cautiously given the large number of gene expression samples for which we could not obtain enzymatic activity measurements for this species: only four *O. faveolata* gene expression samples had enzymatic measurements. However, we did have size and growth measurements for every sample. The light green module was significantly positively associated with buoyant weight change and enriched with terms related to reactive oxygen species metabolism (Fig. S8, Table S3). The green

yellow module was significantly positively associated with tissue growth as increased surface area and contained one enriched GO term: "muscle organ development".

### Symbiont expression profiles with respect to initial size and growth

The average *M. cavernosa* holobiont mapping efficiency was 85.2% with trimmed and filtered reads with per-sample averages of $6.87 \times 10^5$ to *Cladocopium*, and $2.0 \times 10^4$ to *Durusdinium*. Variation in symbiont gene expression profiles were significantly associated with host genet in *Durusdinium* ($p = 0.002$) and marginally associated with weight change in *Cladocopium* ($p = 0.055$). Otherwise, symbiont gene expression was not significantly associated with any other host parameter (Fig. S9).

The *O. faveolata* holobiont mapping efficiency was 81.8% with per-sample average reads of $3.79 \times 10^4$ mapping to *Durusdinium*. Variation in symbiont gene expression was significantly associated with host genet ($p = 0.032$), but not initial weight or weight change over the 4-month recovery period ($p = 0.290$ and $p = 0.513$, respectively) (Fig. S10). We did not identify any significantly differentially expressed symbiont genes from the *O. faveolata* holobiont samples.

## DISCUSSION

### Microfragmentation confers accelerated coral growth with potential tradeoffs

The growth patterns in this study confirms what previous studies (*Forsman et al., 2015*; *Page, Muller & Vaughan, 2018*) and coral hobbyists have already put into practice: microfragmentation confers accelerated coral growth. Additionally, our analyses support the microfragmentation target of ~1 cm² fragments, as this size class may approach the maximum growth benefit while avoiding the increased risk of tissue loss associated with extreme variability in smaller fragment sizes (Fig. S1). Consequently, the growth produced by this technique allows coral nurseries to generate large amounts of coral biomass for outplanting onto degraded reefs in an effort to rebuild these failing ecosystems.

Tradeoffs between growth and factors that promote stress tolerance, such as constitutive immune investment, could limit the long-term success of microfragmentation-based coral restoration. This may occur if microfragmented coral in the nursery cannot withstand stressors experienced after outplanting, as all corals have an ultimate energetic budget that supports growth, metabolism, reproduction, and stress responses (*Lesser, 2013*). The limits at which a coral experiences an energetic threshold that shifts resources between growth and stress responses are not clear.

Some studies have identified positive associations between growth and coral health (*Quigley et al., 2021*; *Wright et al., 2019*) suggesting that, at least within the environmental parameters and time scales investigated, those corals could exhibit rapid growth and withstand stress. Other studies have identified tradeoffs between growth and thermal tolerance (*Cornwell et al., 2021*; *Ladd et al., 2017*) which may be linked to reduced energetics associated with thermotolerant algal symbionts (*Little, van Oppen & Willis, 2004*). Similarly, negative correlations between initial fragment size and immune activity parameters observed here could limit the ability of these fragments to restore

disease-afflicted Caribbean reefs if those differences in immune enzymes (1) persist throughout nursery rearing over longer recovery periods (*e.g.*, 12 months) and (2) actually confer increased susceptibility to disease. Our findings urgently motivate further research to address these concerns by experiments addressing the persistence of microfragmentation trade-offs over longer periods of time and experimental disease challenges. Predation by corallivores can also be a major threat to the long-term success of outplanted corals (*Aeby & Santavy, 2006*; *Gignoux-Wolfsohn, Marks & Vollmer, 2012*; *Page, Muller & Vaughan, 2018*). One study found that up to 27% of outplanted fragments were removed by fish within a single week (*Koval et al., 2020*). That same study found an additional 9% of surviving fragments showed signs of fish predation, which causes physical damage that has been linked to increased susceptibility to diseases such as White and Black Band (*Aeby & Santavy, 2006*; *Gignoux-Wolfsohn, Marks & Vollmer, 2012*; *Page, Muller & Vaughan, 2018*). The disease risk associated with predation wounding further highlights the need to ensure that coral outplants have robust immunity.

In the meantime, there is plenty of reason to have hope for microfragmented corals on Caribbean reefs. A previous report tracked out-planted microfragments over 31 months spanning two bleaching events and found no significant differences in survival between microfragments and larger corals at the same sites (*Page, Muller & Vaughan, 2018*), suggesting that microfragmented corals can withstand bleaching stress while undergoing rapid growth. Mote Marine Laboratory's Elizabeth Moore International Center for Coral Reef Research & Restoration recently reported that outplanted *O. faveolata* microfragments were able to reach sexual maturity, and successfully spawn during the annual broadcast spawning event (*Koch, Muller & Crosby, 2021*). These same fragments had survived a 2015 bleaching event, a Category four Hurricane, and an outbreak of SCTLD (*Koch, Muller & Crosby, 2021*; *Erinn Muller and Coral Restoration Team, 2021*).

Continued research into potential tradeoffs between growth and stress tolerance can inform practices to increase coral growth while limiting detrimental effects on coral health and restoration costs. For example, measuring growth rate and immune capacity throughout a longer microfragmentation recovery period (*e.g.*, 12 months) can help determine the optimal time to outplant. Other aspects of optimizing the microfragmentation procedure should be experimentally explored, documented, and standardized to accelerate the pace of coral restoration knowledge and technology. For example, a recent study found that microfragmented corals grow better on cement plugs than on more expensive ceramic plugs (*Papke et al., 2021*). It is our hope that as we expand upon our depth of knowledge, this promising protocol will continue to evolve to ensure the most promising and resilient fragments will be outplanted on degrading reefs.

## Gene expression reveals biological underpinnings of rapid growth and potential immune tradeoffs

Positive associations between enzymatic activities and initial fragment size provide direct evidence for reduced constitutive immunity in microfragmented corals at the protein level. Gene expression analyses also suggest potential tradeoffs between immunity and growth.

Though they provide less direct evidence, the differentially expressed genes identified in this study present priority candidates for future research investigating mechanisms of immunity and coral growth.

Most of the variation in gene expression was explained by coral genet (ADONIS $r^2$: *M. cavernosa* = 0.25, *O. faveolata* = 0.35), a common finding in coral RNAseq studies (*Parkinson et al., 2015*, *2020*; *Wright et al., 2017*) that requires careful consideration when interpreting differential expression. We take a conservative approach to modeling expression by including many individuals of each genet and by including genet in the DESeq2 model. We observed subtle differences in expression that were associated with initial fragment size and growth over the recovery period. Including these two continuous factors in the model allowed us to isolate genes with linear associations with either initial size (i.e., the extent of microfragmentation) or change in buoyant weight (i.e., growth by calcification). We also conducted a binned analysis comparing expression between "small" and "big" initial size fragments and "fast" and "slow" growing fragments.

Using a continuous scale for our variables, two genes were significantly differentially expressed with respect to growth in *M. cavernosa* samples. The expression of guanylate-binding protein (GBP) was negatively associated with growth rate. GBPs have roles in mediating innate immune responses in multiple types of infections (*Praefcke, 2018*), indicating a possible tradeoff between immunity and growth rate. In *O. faveolata*, several of the 38 genes demonstrating significant negative associations with growth also play a putative role in immunity. For example, cathepsin has been found in multiple immune responses in humans, including the toll-like receptor signaling pathway (*Yadati et al., 2020*) which is also found in innate immune responses in corals (*Skutnik et al., 2020*). Integumentary mucin C.1 plays a role in microbial infection defense in mucus (*Bakshani et al., 2018*). The mucus layer is the first layer of defense in corals, and thus a critical component of a coral's tolerance to infection.

WGCNA identified groups of co-regulated genes associated with growth and immune phenotypes. Opposing correlations between module expression with growth metrics and immune parameters reflect potential gene expression tradeoffs. The three "tradeoff" modules identified by WGCNA in *M. cavernosa* are enriched with biological activities related to growth and skeletal development. A recent study identified ossification-related processes enriched in *Stylophora pistillata* exposed to reduced pH (*Scucchia et al., 2021*). That study found that acidification-resistant corals demonstrated high expression levels of cell adhesion genes, similar to the enrichment of cell junction and cell migration terms identified in the dark magenta module associated with calcification here in *M. cavernosa*. We did not observe similar enriched functions associated with growth in *O. faveolata*. Only one module associated with calcification contained any significantly enriched GO terms: light green. This module contained 69 significantly enriched GO terms with descriptions including responses to stress and reactive oxygen metabolism. The dissimilarity in enriched functions between these two coral species may reflect differences in their microfragmentation-mediated growth responses.

The associations between buoyant weight change and gene expression in *O. faveolata* offer an opportunity to reveal biological mechanisms underlying calcification.

The fragments that exhibited the most growth also had significantly higher abundances of transcripts for glutamic-rich proteins, which are key modulators of biomineralization across taxa (*Gorski, 1992*) that were recently identified in association with collagen within the skeleton of *Stylophora pistillata* (*Mummadisetti, Drake & Falkowski, 2021*). Transcripts encoding carbonic anhydrases, enzymes that catalyze the interconversion of $CO_2$ to bicarbonate ions driving coral calcification, exhibited the highest expression among fragments with low–intermediate growth (Fig. 5). This counterintuitive finding warrants further investigation into the mechanism of carbonic anhydrase-driven calcification, especially as previous work has shown that environmental factors can affect enzymatic activity (*Zoccola et al., 2016*). The unannotated genes identified in this study represent further opportunities to explore regulators of coral calcification.

Differentially expressed *O. faveolata* genes in the binned analysis were mostly uncharacterized, prompting further investigation of growth-related genes in this species. In the binned *M. cavernosa* analysis, a transcript with homology to tachylectin-2 was upregulated in smaller fragments. This gene product has a putative role in microbial recognition and agglutination in Japanese horseshoe crabs, *Acropora*, *Montastrea*, and *Nematostella* species (*Hayes, Eytan & Hellberg, 2010*). Several transcripts related to biomineralization were upregulated in smaller fragments, providing molecular insight into mechanisms of microfragmentation-mediated growth. The calcium ion channel polycystin-2, and other proteins in its family, have been identified as members of the skeletal organic matrix in multiple coral species (*Zaquin et al., 2021*). Both transmembrane protease serine 9-like, and coadhesin proteins, have both been linked to calcifying processes and have been found upregulated in corals with high calcification rates (*Kelley et al., 2021*; *Mummadisetti, Drake & Falkowski, 2021*; *Peled et al., 2020*).

We sampled for gene expression after the 4-month recovery period, so these results only reflect biological differences that persist long after fragmentation. This sampling time was intentional as we did not want to capture stress responses to the necessary wound inflicted during fragmentation, though future experiments may include more sampling timepoints (such as at the time of outplanting size, ~3 cm$^2$) to better capture the biology of wound repair and subsequent growth. We only removed tissue from the edge of each fragment to prepare the gene expression libraries, and we airbrushed the remaining edge and center tissue to measure enzymatic activities. Thus, this study only reflects gene expression at the edge of the growing fragments and cannot reveal potential difference in immune activity of edge relative to center. Given that smaller coral fragments have a higher ratio of edge:center tissue than larger fragments, differences in biological activities between these two types of tissue could underlie differences in growth and survival.

## Associations between symbiont characteristics and growth

This study revealed an abundance of reads mapping to *Durusdinium* in *M. cavernosa* and *O. faveolata* (Fig. 3). Previous studies indicate that these coral species are typically dominated by *Cladocopium* in the Caribbean (*Drury et al., 2020*; *Serrano et al., 2014*; *Sturm et al., 2020*; *Warner et al., 2006*), with few exceptions (*Manzello et al., 2019*). *O. faveolata* fragments are dominated by *Durusdinium* at this particular sampling location, perhaps

because of the original environment from which the broodstock was collected or due to environmental parameters of the land-based system that promote that particular algal type. Symbiont shuffling can promote *Durusdinium* dominance in a previously *Cladocopium*-hosting coral after recovery from thermal stress (*Cunning & Baker, 2020*). Given that the suspected thermotolerance conferred by *Durusdinium* relative to *Cladocopium* may have energetic costs (*Jones & Berkelmans, 2011*) that manifest as reduced growth rates (*Cunning et al., 2015*), reef managers should continue to monitor shifts in dominant symbiont types. Comparing growth rates across corals hosting different symbiont types was not a planned goal of this research, but the observed differences in algal symbiont proportions among *M. cavernosa* samples did allow us to conduct a limited investigation. We did not find the association between growth and *Durusdinium* that others have reported, but these results are constrained by our limited sample size.

We did not find any algal symbiont genes significantly associated with coral growth. Other studies have also seen transcriptional stability within the algal symbiont (*Barshis et al., 2014*; *Davies et al., 2018*; *Leggat et al., 2011*), which may be a result of host buffering or a consequence of the unique genomic organization of the symbionts. Dinoflagellate chromosomes lack histones that regulate transcriptional dynamics in their hosts (*Rodriguez-Casariego et al., 2018*) and exist in a condensed liquid crystalline state arranged into topological domains (*Marinov et al., 2021*). Future studies could examine symbiont responses at the level of protein or metabolite rather than gene expression in order to reveal associations between algal biology and host phenotypes.

## CONCLUSIONS

This study provides further evidence that the microfragmentation technique promotes faster coral growth. Future studies should continue to investigate the association between size and growth rate in corals, specifically how the surface area:edge ratio may influence growth dynamics. Enzymatic activities and specific gene expression associations with initial fragment size and growth rate reveals a possible tradeoff between microfragmentation and immunity after a 4-month recovery period. However, restoration efforts report great success with outplanted microfragments in Caribbean reefs. Future studies could investigate whether these reductions in immune activity observed in this study persist throughout the recovery period and translate to increased disease susceptibility. These gene expression patterns associated with coral calcification highlight the role of glutamic-rich proteins in biomineralization. This study identified currently unannotated genes as potentially important drivers of coral growth that represent key opportunities for further molecular characterization.

## ACKNOWLEDGEMENTS

We are grateful to Erich Bartels for collecting the coral samples under FKNMS permit number FKNMS-2015-130 and to Burt D. Snover for field assistance. We thank the Texas Advanced Computing Center and Smith College Computing and Technical Services for computational resources.

### Funding

Funding was provided by a Texas State Aquarium grant awarded to Mikhail Matz. The funders had no role in study design, data collection and analysis, decision to publish, or preparation of the manuscript.

### Grant Disclosures

The following grant information was disclosed by the authors:
Texas State Aquarium grant awarded to Mikhail Matz.

### Competing Interests

The authors declare that they have no competing interests.

### Author Contributions

- Louis Schlecker analyzed the data, prepared figures and/or tables, authored or reviewed drafts of the paper, and approved the final draft.
- Christopher Page performed the experiments, authored or reviewed drafts of the paper, and approved the final draft.
- Mikhail Matz conceived and designed the experiments, performed the experiments, authored or reviewed drafts of the paper, and approved the final draft.
- Rachel M. Wright conceived and designed the experiments, performed the experiments, analyzed the data, prepared figures and/or tables, authored or reviewed drafts of the paper, and approved the final draft.

### Field Study Permissions

The following information was supplied relating to field study approvals (*i.e.*, approving body and any reference numbers):

Field collections were collected at Florida Keys National Marine Sanctuary (FKNMS) under permit number FKNMS-2015-130

### DNA Deposition

The following information was supplied regarding the deposition of DNA sequences:
The sequences are available at GenBank: PRJNA764071.

### Data Availability

The raw data (growth measurements, count data for both coral species M.cavernosa and O.faveolata) is available at GitHub:
https://github.com/rachelwright8/Microfrag_Growth_Immunity_TagSeq.

### Supplemental Information

Supplemental information for this article can be found online at http://dx.doi.org/10.7717/peerj.13158#supplemental-information.

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
