# Peer review of "Mechanisms and potential immune tradeoffs of accelerated coral growth induced by microfragmentation"

_PeerJ, doi:10.7717/peerj.13158_

## Round 0.1 · original submission · Major Revisions

Three expert reviewers have evaluated your manuscript and provided thorough reviews. This is an important piece of research, however there are a number of issues that need to be attended to in a resubmission.

Reviewer 1 ·

Basic reporting

I was very excited to read this paper as this is an important study and finding these trade-offs is really furthers our understanding of coral immunity. The use of Mote micro-frags is a great idea and has the potential to produce some very interesting data. I feel this paper could be improved by going a bit deeper into the data analysis and tightening up the main findings. The title itself is compelling but the data, as presented, don’t really tell the same story. I would love to see more hypotheses presented and tested.

Introduction
The last paragraph of the introduction could be a bit clearer on what the motivation of this study is. Is it to find the mechanisms that are allowing for faster growth in the microfragments? Or is it looking for trade-offs with growth? As I see it, this study isn’t really about the process of microfragmentation, but more looking about using this set-up as an experimental platform to study the rapid growth of microfrags and the potential trade-offs.

Results
Line 246 – seems like discussion or potentially a justification of methods that could be moved into the methods.
Why are growth rates of the coral fragments not presented? I would like to the range of growth rates and how many frags of each species had a faster vs slower growth rate.
Figure 1 – can you highlight or bold the stats that are significant or simplify the statistical output. Would an average reader understand what each of the pMCMC values mean? How do I interpret this and understand what interaction was significant?

Line 256 – “Immunity results indicate some tradeoffs between enzymatic activity and growth “ this seems like a discussion point, please give stats or this negative correlation.

Figure 2 – again can you add information to help the reader understand what is important in these graphs? Which have significant correlations between growth and immune paraments? Bold and star the significant interactions.

Also, why is log initial weight used instead of growth rate? I understand that smaller frags grow faster but its not that convincing without using the actual fragment growth rates. Also, this is the title of your paper. Another idea is to bin the fragments into groups that all are similar growth rates and show the immune activity of those. Also perhaps grouping each fragment into fast, medium and slow growth rates and showing the immune properties/symbionts and gene expression of those would illustrate the points well too.

Figure 3 and the results of the symbiont identification could be elevated a bit. Was there different growth rates between the individuals that different symbiont populations. For example there were several M. cavernosa that didn’t have Durusdinium, what was their growth rates? Was there a significant correlation between % of any symbiont type and growth rate? Did symbiont population drive growth in any way? I think there are numerous ways to test this in your data set. Even grouping the individual stacked plots presented in figure 3 by slowest to fastest growth would make this figure more interesting.


Figure 4 and gene expression - Is DESeq the best way to get the patterns of this data if you only have a continuous variable? Why wouldn’t you bin them in groups of low, medium and high growth rates to look for DEGs or use WGCNA which is made for this type of analysis. I feel that the gene expression data could be much deeper and give more in depth results. There also isn’t an appreciation for the immune genes or GO terms that can be easily seen. Again, this is the title of the paper? Shouldn’t there have been an effort to add this trade off hypothesis to the figure? Also looking for trade-offs with the gene expression would be good, like were there any growth GO terms that were upregulated and immune processes downregulated or vise versa?

Where is Mcav gene expression data? DEGs are not the most important thing in this type of data set. I feel a lot of data was left on the cutting room floor and not analyzed completely.

I also really appreciate that you presented the uncharacterized genes, as this is a common issue with non-model organism, but are there are ways to get some basic information about these gene pFam or even eggnog annotation that will you the domains of the base ortholog? Perhaps also putting them is separate figure would help direct the focus of the gene expression figure to the known genes and allow for some deeper/different analysis of the uncharacterized genes.

The heat map could be more informational and impactful and could be a valuable contribution to the genes involved in growth rate. Some grouping by GO term or function would be great. Show the immune genes clearly. Also the legend is backwards from conventional heat maps.

Perhaps can you regress the gene expression to the immune parameters? These are additional phenotypes to help pull out patterns. Then you could overlay the growth data to really show nice correlation and trade-offs.

Discussion
This discussion could use some re-organizing. for example the entire paragraph on line 282 seems out of place. discussing the data as far as the trade offs in the smallest fragments and as far as growth rate of the fragments seem like two different questions to me.

Experimental design

see above

Validity of the findings

see above

·

Basic reporting

All appropriate data and scripts are reported in the text, available online, and any sequence projects submitted to the proper databases.

Figures.
S2. It is unclear whether these are all the fragments produced from one colony or a subset without a caption. Also, not sure what the arrows represent.

S3&S4. Perhaps add Ofav into the title so readers know which species the cluster analysis is for.

S5 does not have ‘A’ or ‘B’ as panel titles. Would also prefer to see a genet pvalue somewhere on the figure (and in S6), but would be redundant if in both panels, so maybe report in the caption or just in panel A?

Text.
Lines 54 & 65. Needs space after the semicolon in citations.

Lines 75-77. Unclear what success means in this context, survival, increased growth after outplanting?

Lines 78-83. This is the crux of the study and could be expanded upon with similar studies investigating growth rates and potential tradeoffs. See Edmunds 2017 (Ecology) and Denis et al. 2011 (MEPS) and 2013 (PLoS); although these investigate growth tradeoffs correlated with environmental stress, there are relevant implications of fast growth/wound repair under stressors that can be applied here.

Line 120-121. “the same protocols on 12 Mar 2016” reads like you used the same protocol from that date, not the start date 10 Nov.

Line 154. Peroxidase activity is given the acronym PO but is defined/used as POX in the introduction and in the discussion.

Line 255. As in a sufficient amount? What does “high-quality tissue” mean?

Experimental design

Line 108. The “successively smaller fragments” make it difficult to glean relative fragment size. Did you have relative size bins? What is the sample size per fragment size per genet per species? Also, should refer to the supplemental figure here.

Line 126. Although the methods and analyses cannot be re-done, should take into consideration in the discussion that you airbrushed from the entire fragment, but gene expression was from the growing edge of fragments. Do you think you’d find different immune activity if you only airbrushed from the growing edge as well? Larger fragments would, in theory, have a smaller edge area so potentially reduced enzymatic activity (or dampened activity due to the presence of more ‘center’ tissue) driven by the rest of tissue instead of just the growing edge?

Line 200. In the same vein as above, could the association between growth and area of growing edge relative to total surface area (instead of or in addition to) be an additional factor influencing growth rate?

Validity of the findings

Line 231. Again, hard to understand the relevance of findings without knowing sample size, even if relative.

Line 242-243. A great demonstration of how growth, especially in smaller sizes, can be so variable.

Line 259. The sentence is slightly misleading when reviewing the figure. Although it is the only enzyme activity that was not significantly different from coral size, stating “declines in activity” is counterintuitive to the figure and the findings. Larger fragments exhibited higher activities which are reflected in the panels, so why would activity decline with coral size? Or did you mean to say growth rate, if smaller fragments grow more, they have lower enzymatic activity. Please clarify.

Line 340. Stated that “target of ~1cm2 fragments” but readers did not know until now that you had fragments that small based on experimental design.

Paragraph 351-356. Weak in discussing the tradeoffs, as enzymatic activity and any immune-related transcripts are not discussed here. Tie in these results for comparison and support of statement or combine with the following paragraph to provide evidence-based research for both positive and negative associations.

Lines 374-376. Is any of this related to the actual size of outplants, or attributed to nursery conditions (husbandry effects, like ‘healthier’ and ‘yummier’ corals), reef site (more corallivores than others), or an effect of time since outplant? How do these studies relate to your findings?

Line 386. Do they still grow as ‘rapidly’ (or more so) on the reef vs. nursery? This has implications on whether to grow out in land-based versus field nursery if growth rates are indeed more rapid in one setting versus another.

Paragraph 438-445. Previous studies are documenting wild coral symbiont types, and it is likely that these corals could harbor different symbiont types on land using a different water source than reef water. The argument for post-stress shuffling isn’t clear unless the shift came about as a stress response to microfragmenting (unlikely). What is more interesting is Durusdinium typically leads to lower growth rates (Cunning et al. 2015, Coral Reefs), and you didn’t see significant differences between the two species as M. cavernosa harbored both Cladocopium and Durusdinium.

Additional comments

This study is a great foundation for what we know and do not know about the potential tradeoffs associated with the microfragmentation process. Able to generate sufficiently sized outplants much faster than using larger fragments, restoration practitioners can upscale to potentially replace the biomass that’s being lost. The reproducibility of results regarding smaller sizes grow faster is important to the restoration field, but the group also introduced applicable future experiments to further understand the potential repercussions of a corals energy budget. For example, it would be worthwhile to track immune activity and gene expression along the growing edge alone (not the entire fragment) at multiple time points, as it is likely the wound healing response occurs within the first month of lesion/fragmenting (see Denis et al. 2013, PLoS) and it could be possible that the immune activity was dampened by looking at the whole fragment instead of just the growing edge. Moreover, coupling the experiment with a disease exposure component could elucidate the potential immunity tradeoff further and provide critical implications for direct outplanting post-fragmentation.

·

Basic reporting

Overview:
Schlecker et al. conducted a manipulative study that aimed to quantify the growth rate, immune protein, and gene expression of Orbicella faveolata and Montastraea cavernosa four months after being fragmented into varying sizes. The authors conclude that smaller fragments grow faster, but there is a negative association between fragment size and immune competence suggesting that there may be tradeoffs between accelerated growth and immunity. Overall, the paper is well written and poses timely questions, which are both highly relevant and novel. I thoroughly hope that this group continues to ask and answer important questions, such as those posed within this paper, in collaboration with groups conducting boots on the ground coral restoration.
One challenge I had with the paper though was the presentation and tone suggests that coral restoration practitioners haven’t considered potential tradeoffs or the threats like disease on corals created through microfragmentation. However, this practice has been going on for almost a decade and although not widely published, the overall survival rates are quite high even when outplanting within the endemic areas of SCTLD (see references listed below). Placing the paper within the current precautionary tone ignores about a decade’s worth of that information. I suggest that this paper should not be focused as a precautionary tail to beware when outplanting microfragments; perhaps rephrase the message to include restoration practitioners’ common knowledge that “there may be a reduced immune competency associated with microfragmentation, yet these corals are still surviving after outplanting. How/Why is that happening?!”.
Also, the important implications of the present study could help guide further care when corals are in nursery settings or guide when corals should be outplanted to increase survival. For example, Orbicella faveolata microfrags are typically grown for 8 months prior to outplanting and M. cavernosa is grown for 12 months or more. What does the immune status look like at the time of outplanting? Additionally, corals are not outplanted until they completely cover the plug. It is difficult to tell, but only a few of the corals in the photograph from March 2016 would actually have been ‘outplantable sized’. Does this reduced immune competence found in the study further support the long grow out period current practitioners use?
One caveat I am also struggling with…how do we know that the differences in immune related activity are associated with fragmentation process and not simply related to the size of the corals? For example small sized corals (whether fragmented or not) may have reduced immune capacity. We don’t have a control in this study so perhaps there are comparative measurements from other studies? What do we know about the measurements in the present study and those taken from ‘wild’ corals of comparable sizes or just any ‘wild’ corals in general? A comparison to field collected samples would be really interesting component to add to the discussion.

Additional References to Consider Including or reading for context:
White paper to consider integrating (and also cite instead of E. Muller personal observation Line 388):
https://floridadep.gov/rcp/coral/documents/can-sctld-susceptible-species-be-outplanted-florida%E2%80%99s-coral-reef-acceptable

Another resource on outplanting survival of microfragments (incorporating predation effects):
Smith, Kylie M., Devon M. Pharo, Colin P. Shea, Brian A. Reckenbeil, Kerry E. Maxwell, and C. Sharp. "Recovery from finfish predation on newly outplanted boulder coral colonies on three reefs in the Florida Keys." Bulletin of Marine Science 97, no. 2 (2021): 337-350.

We also showed that microfragmented and fused corals can spawn a few years after outplanting: https://www.the-scientist.com/features/restored-corals-spawn-hope-for-reefs-worldwide-68368
I think the references listed above (although I realize not all peer reviewed) should be included if possible and help guide a more balance discussion.

Experimental design

Abstract:
I think the gene expression results are specific to only one species, Orbicella faveolata, please make that clear in lines 39 – 42, or ignore me if I am incorrect.
Lines 43 – 45: I think this sentence could be broader. SCTLD is just the latest disease threat to the Florida Reef Tract and all of the other diseases certainly haven’t gone away. Disease has been a major issue restoration practitioners have had to consider since the beginning of restoration within this region.
Lines 47 – 48: revisit the concluding sentence after reconsidering broad presentation of the paper.

Introduction:
Line 63: I’m not sure Dobbelaere et al. 2020 is the best reference here since it focuses on FL hydrodynamics and particle modeling, perhaps include Meilling et al from USVI or Sharp et al. 2020, Williams et al. 2021 from FL
Line 66: several other studies showed antibiotics work to treat SCTLD (Neely et al. 2020 PeerJ, Aeby et al. 2019)
General: Recommend including a paragraph on gene expression association with growth/immune competency in corals. There is a section on immune-related proteins but not gene expression.
Line 75 – 76: seems to be a word missing in this sentence; also, what does ‘success’ mean here?
Line 96 – 98: I would re-write this to include the use of gene expression as a tool for characterizing both growth and immune competency since that is what you actually use for conclusions.
Methods:
Line 104: provide more details of the coral nursery…this was likely the NOAA Rescue Nursery, where corals were removed from construction projects (seawalls etc – extremely shallow areas) and preserved in the nursery for research purposes instead of being lost in the demolition process.
Line 108: please explain how you determined the size the fragment the colonies. Looking at the size distributions there were many more around the 1cm mark compared with the larger fragments and Figure 1A appears to have two ‘size classes’ rather than an even distribution.
Line 113: How was the water treated and filtered? I would re-word this sentence to be more informative. High light and low turbidity are subjective terms.
Line 116: usually it’s corrugated clear PVC, not metal. If you used metal, why would additional shade cloth be needed – this part is confusing.
Line 117: when was PAR measured? How often?
Line 119: I imagine there was physical mitigation needed to remove algae from the plugs, otherwise, your discussion of the high amount of attention needed to grow microfragments (lines 347-348) doesn’t make sense (if snails and siphoning is all that was needed for 4 months).
General: what was the temperature and pH of the water?
General: remain consistent using past tense (examples: line 217 remain should be remained, line 232 includes should be included, line 283 and 284 ‘is’ should be ‘was’, 298 and 299 ‘have’ should be ‘had’ etc.)

Validity of the findings

Results:
Line 231: separate the number of fragments by species (x ofav and x mcav)
Line 236: ‘larger fragments grew less’, since you are basically doing linear models, I would think it would be more appropriate to say that there was a positive or negative association rather than saying ‘larger fragments grew less than smaller fragments’, or you could say that size was a predictive factor associated with growth. As it states now, I can’t tell what ‘large’ vs ‘small’ is since size is treated as a continuous variable.
Line 255: how many from each species?
Line 257: see same comment for line 236
Line 260: change ‘across’ to ‘between’ (between is appropriate when comparing two things, here two species)
Line 270: I recommend adding/altering the text ‘which was not statistically significant’, nor did we observe…’ The data suggesting there may be a negative trend in growth for Mcav isn’t compelling here so I am not sure alluding to it twice in one sentence is necessary or appropriate.
Line 280: differentially expressed when comparing what?
Line 315 – 318: this sentence keeps tripping me up. I suggest rewording it for easier interpretation.

Discussion:
Line 345: what does ‘processing time’ mean?
Line 347 – 349: there is a lot more to consider when thinking about many small fragments vs fewer larger fragments than just algae mitigation. It is more difficult to maintain the health of large corals within controlled land-based systems (need more space, water flow, electricity etc), there are more issues with pest interactions when there is a lot of skeleton within the colony and once an issue arises the entire colony could be at risk. Also, where are you going to get the biomass to have these large corals? Having many small corals allows for redundancy of genetics within the broodstock, limits the amount of initial biomass necessary, and is easier to maintain for a healthy outplant. By outplanting clusters of small fragments you create the large coral on the reef within a few months to a year, rather than within a land-based system (which costs a lot more to maintain than an outplant on a reef).
Line 370 – 373: Stating that restoration can do more harm than good (and the references listed in this statement that focus on grounding sites, destroyed reefs from quarry excavation, and moving fragments from one reef to another) are examples of transplantation projects that are substantially different than what most restoration organizations do today – particularly the ones on Florida using microfragmentation for restoration. For example, all of Mote’s broodstock was originally acquired through the NOAA Rescue Nursery (corals that were removed from seawalls ahead of construction projects that would have killed them otherwise) or through corals of opportunity, which is defined as a coral detached from the reef that is in a location where it would likely die if left alone (i.e., in sand). New stock is acquired through assisted sexual reproduction events that would not have been successful without human intervention. I think perpetuating falsehoods about restoration causing more harm than good leads to continued misunderstandings of the practices. I recommend lines 370 to 374 get deleted or are rewritten to accurately reflect best practices of today.
Line 408 – 409: here you should also mention that quantifying the microfragments when they are the size of outplants (3cm diameter or more) is another area of attention in future studies.
Line 411: two genes were differentially expressed…in what way? Remind reader if it was up or down in respect to growth in Mcav. Same in line 416.
Line 415: I suggest making this sentence more tempered “The down-regulation of the GBP gene in M. cavernosa fragments exhibiting higher growth suggests further investigations into the potential for immune tradeoffs among faster growing corals is warranted.”
Line 431: change ‘urges’ to ‘suggests’
Line 441 – 445: actually we know that the majority of our Ofav (like 99%) are dominated by Durusdinium at Mote. This could be because our original broodstock were collected from extreme environments (seawalls) or because our land-based system promotes it.

Conclusion:
Consider toning down lines 461 – 463 based on my comments above.

Additional comments

No Comment

---

## Round 0.2 · Minor Revisions

Two expert reviewers have evaluated your revised manuscript and both commend you on the effort you put in to improve the manuscript. There are some minor, but nevertheless important, comments that have been made on this resubmission. Please ensure that you consider all of these when preparing a new version of the manuscipt.

Reviewer 1 ·

Basic reporting

I appreciate the work the authors did to make the points of this manuscript clearer. The figures are improved and the supplemental figures address many of my issues and give the reader more data to consider. I support publication of this new version.

Experimental design

No comment

Validity of the findings

This paper will be a great resource to reef managers who conduct fragmentation and restoration as well as basic scientists interested in immune trade-offs and gene expression patterns related to growth. very nice!

·

Basic reporting

The authors did a great job addressing all of my initial comments, suggestions, and concerns. My only non-editorial suggestion is that I think identifying if and when this tradeoff between growth and immune parameters subside could be very useful to guide WHEN to move the corals from the 'safety' of a nursery to an outplant site. This could be added as a sentence or two within one of the relevant Discussion paragraphs. Otherwise, I have only editorial comments to suggest at this point. Thanks to them for the significant effort on this paper.

Abstract:
Suggest changing Line 9 to the following: Microfragmentation is the act of cutting corals into small pieces (~1 cm2) to accelerate the growth rates of corals when compared to maintaining larger-sized fragments.

Line 13: change to "Here we compared..."
Line 14: add comma after 'corals'
Lines 19-20: suggest changing to "Innate immunity enzyme activity assays and gene expression results suggest a potential tradeoff after microfragmentation between growth rate and immune investment.
Line 24: suggest changing to something like "...microfragmentation recovery period that may affect growout survival and disease transmission after outplanting".

Intro
I realize the added text associated with the history of disease influencing the current state of the Florida reef tract was in response to my initial review, but the text sort of over simplifies the situation now. The state of FL reefs now, and much of the Caribbean (as noted) is definitely in part due to disease, but also major bleaching events and hurricanes. I think that integrating a balance into these statements is necessary to be accurate (lines 34 - 37 primarily).

Lines 43 - 46: adding antibiotics to sick coral is not a form of 'restoration', it is generally considered a form of 'intervention' to prevent loss. Separate the intervention from the restoration text within this sentence to clarify and better wrap up this paragraph.

Lines 79-82: this sentence structure is a bit awkward, maybe rethink it

Line 95: change to "The present study..."

Methods:
Line 223: change to "We used..."

Results:
LIne 263: change to "...expansion and as change in buoyant weight, which included..."

Lines 264 - 266: are these two averages and SD related to the 'small' and 'big' frags for each species? Clarify.

Discussion
Line 459: I think you could just use Koch et al. 2021 here too

Experimental design

No comment

Validity of the findings

No comment

Additional comments

No comment

---

## Round 0.3 · accepted · Accept

I am satisfied with the changes that have been made to the manuscript.